# A Nitrogen-Rich DOPO-Based Derivate for Increasing Fire Resistance of Epoxy Resin with Comparable Transparency

**DOI:** 10.3390/ma16020519

**Published:** 2023-01-05

**Authors:** Jiayi Lu, Boyu Cai, Wendi Xu, Luze Wang, Zhonglin Luo, Biaobing Wang

**Affiliations:** Jiangsu Key Laboratory of Environmentally Friendly Polymeric Materials, School of Materials Science and Engineering, Jiangsu Collaborative Innovation Center of Photovoltaic Science and Engineering, Changzhou University, Changzhou 213164, China

**Keywords:** epoxy resin, flame-retardant mechanism, fire resistance, transparency

## Abstract

To endow synergistically epoxy resin (EP) with excellent fire resistance and high optical transparency, a nitrogen-rich DOPO-based derivate (named as FATP) was synthesized and incorporated into EP. It showed that the incorporation of the FATP reduced the fire hazard of the EP, as demonstrated by the fact that the EP/4% FATP blends gained a UL-94 V-0 rating and an LOI value of 35%, with the lowest values of the THR (86.7 MJ/m^2^), the PHRR (1059.3 kW/m^2^), and the TSP (89.6 MJ/m^2^). The presence of the FATP also reduced the thermal stability and the crosslinking density whilst improving the curing reaction and the storage modulus of the EP/FATP blends. The TG-FTIR spectra showed that •HPO/•PO free radicals and some nonflammable gases (HN_3_ and NH_3_) were produced during the pyrolysis, and the characterization (SEM, Raman spectroscopy, and XPS) of char residues confirmed that the FATP facilitated the formation of continuous and compact carbon layers of greater graphitization degree. It was thus concluded that the FATP played the flame-retardant roles in both the gas and condensed phases. Furthermore, the FREPs kept almost identical transparency as the pristine EP, and mechanical properties were also slightly enhanced. The FREPs presented in this work show promising applications in the fields of advanced optical technology.

## 1. Introduction

Epoxy resin (EP), one of the most used thermosets, has been an indispensable basic material for high-end manufacturing industries such as 5G communications and rail transit, thanks to its excellent comprehensive performances including thermal stability, mechanic strength, chemical resistance, and transparency. However, like many other petroleum-based polymer matrices, the EP also suffers from intrinsic flammability with serious melt-dripping because of the organic characteristics, which restricts the broad application in the fields requiring harsh fire safety [1,2]. It is well-known that fires as one of common accidents threaten seriously human security and the environment and bring about annual costs of tens of billions of dollars globally [3]. Thus, endowing EP with better fire resistance becomes urgent from both the perspectives of expanding its application and sociological pressure. In the past decades, halogenated flame retardants (FRs) have been used extensively to enhance the fire resistance of EP due to their inexpensiveness and better flame-retardant efficiency. Unfortunately, halogenated FRs release some toxic gases such as halogenated dioxins and dibenzofurans during the combustion [4,5], and thus, a majority of halogenated flame retardants are forbidden in order to abide by RoHS. Therefore, it is an inexorable trend to develop green FRs to meet with the regulations about environmental protection and human health [6,7].

Heretofore, many efforts have been made to prepare halogen-free FRs which could enhance dramatically the fire resistance of the polymer matrix. A lot of works have demonstrated that the incorporation of phosphorus- [8,9], nitrogen- [10,11], silicon- [12,13], and sulfur-containing [14,15] compounds into EP is an effective method to increase the flame retardancy. Among these compounds, phosphorus-containing compounds have been mostly used for their low toxicity and high flame-retardant efficiency [16]. Lately, 9,10-dihydro-9-oxa-10-phosphaphenanthrene-10-oxide (DOPO) has become a common phosphorus-containing flame retardant for EP due to its better stability and inherent fire retardancy [17]. Nevertheless, it suffers limitations that UL-94 V-1 rating is achieved only at its high loading level when used alone [18]. Luckily, there is a reactive P−H bond in the DOPO, other flame-retardant elements (silicon [19,20], sulfur [21,22], nitrogen [23], and boron [24,25]) or groups (imidazole [26,27], triazole [28,29], and triazine [30,31]) can be integrated into the DOPO through the reaction between the P−H bond and unsaturated bonds including double bonds and epoxy groups, and many DOPO-based derivatives with high flame-retardant efficiency are thus prepared. It is worthwhile pointing out, however, that the introduction of additive FRs inevitably deteriorate mechanical performances or transparency [32]. Especially, the optical transmittance is a vital parameter for EP using in the fields such as LED and transparent coatings; thus, a lot of works have been conducted to prepare flame-retardant EP resins (FREPs) with better transparency [33,34,35,36]. Wang et al. [37] reported that the addition of a 10 wt.% phosphorus/nitrogen-containing derivative (DPAP) endows EP with a V-0 rating and an LOI of 35.2%, but its transmittance decreases obviously from 87.8% of the pure EP to 72%. Despite some achievements were obtained, it is still a research hotspot to simultaneously endow EP with outstanding flame retardancy and high transparency.

As we all know, nitrogen-containing heterocycles, such as tetrazole, have the nature of thermal stability and flame retardancy [38]. Particularly, 5-amino-1H-tetrazole (ATZ) is a stable nitrogen-rich heterocycle with unique four nitrogen atoms and thus is considered as a greatest potential material to fabricate a variety of high-efficient flame retardants [39]. Additionally, 2-furaldehyde is a bio-based compound and can promote powerfully char-forming capacity [40,41]. Thus, in this work, a nitrogen-rich DOPO-based compound (FATP) was synthesized using 2-furaldehyde, 5-amino-1H-tetrazole, and DOPO as raw materials. Subsequently, FREPs with different FATP loading levels were prepared using the 4,4-Diaminodiphenylmethane (DDM) as a curing agent. The effect of the FATP on the curing behavior, flame retardancy, thermal stability, combustion behavior, mechanical properties, and transparency of FREPs was investigated in detail. Finally, the flame-retardant mechanism was proposed through the analysis of pyrolysis gaseous products and char residues.

## 2. Experimental

### 2.1. Synthesis of the FATP

The synthesis route of the FATP by the one-pot method is illustrated in Figure 1. The detail experimental process is described as follows. ATZ (3.403 g, 0.04 mol) and 250 mL absolute alcohol were feed into a 500 mL round three-necked flask connected with a magnetic stirrer, a reflux condenser, and a constant pressure dropping funnel. After the complete dissolution of ATZ at 50 °C, 2-furaldehyde (3.843 g, 0.04 mol) in absolute alcohol (20 mL) was added dropwise through the constant pressure dropping funnel. Subsequently, the mixture was stirred for 12 h at 75 °C, and then, DOPO (8.64 g, 0.04 mol) was added into the above mixture, followed by stirring for another 12 h. Finally, the solution was cooled to room temperature, and the solid precipitate was then filtered, washed with absolute alcohol three times and dried in a vacuum oven at 80 °C overnight to give 10.76 g of FATP as white powder (71% yield).

### 2.2. Preparation of Flame-Retardant Epoxy Thermosets (FREPs)

Firstly, the predetermined FATP was added into the preheated EP at 160 °C and was magnetically stirred to obtain a homogeneous and transparent solution. Then, after the above solution was cooled to 90 °C, the DDM was added and stirred to dissolve completely. Finally, the mixture was injected rapidly into a preheated silicone rubber mold, and the curing process was performed in an oven at 100 °C for 2 h and 150 °C for 3 h. The specific formulas of the pure EP and the FREPs are listed in Table 1.

## 3. Results and Discussion

### 3.1. Characterization of the FATP

The spectra of FTIR, ^1^H NMR, and ^31^P NMR are shown in Appendix A. In the FTIR spectrum of the FATP, some characteristic peaks appeared at 1640 cm^−1^ (υ_C=N_), 1244 cm^−1^ (υ_P=O_), 1215 cm^−1^ (υ_P−O−Car_), and 750 cm^−1^ (υ_P−C_). It was noteworthy that triple absorption peaks at 3150–3465 cm^−1^ ascribed to υ_-NH2_ of ATZ were merged into a single peak at 3432 cm^−1^ (σ_NH_) for the FATP. The disappearance of the stretching vibration peaks at 2435 (σ_C=N_, DOPO) and 1691 cm^−1^ (σ_CHO_, FA) revealed the consumption of DOPO and FA. As seen in Appendix A, the signals at 6.35–6.45 ppm were assigned to the proton of −NH−, the multiple signals at 7.00–7.86 and 8.15–8.37 ppm corresponded to the protons of aromatic hydrogens and heterocycle, respectively, and the ones at 5.36–5.45 ppm are due to the resonance of the aliphatic C−H bond. Furthermore, two chemical shifts appeared at 27.01 ppm and 27.75 ppm in the ^31^P NMR spectrum of the FATP due to the existence of chiral carbon, which was connected with the phosphorus. It can be inferenced that the target product FATP was synthesized successfully according to the above test results.

### 3.2. Curing Behaviors of FREPs

Figure 1 presents the non-isothermal DSC curves as well as the linear fitting curves of the ln(β/T_p_^2^) and ln(β) against 1/T_p_ following the Kissinger (Equation (1)) and the Ozawa (Equation (2)) methods [42,43]. The reaction activation energy (E_a_) was obtained from the slope of the fitted curves, as listed in Table 2.
(1)$$\ln\frac{\beta }{T_{p}^{2}}=\ln\frac{\mathrm{AR}}{E_{a}}-\frac{E_{a}}{{\mathrm{RT}}_{p}}$$
(2)$$\ln\beta =\ln\frac{{\mathrm{AE}}_{a}}{R}-\frac{1.052E_{a}}{{\mathrm{RT}}_{p}}-5.331$$
where in β, T_p_, A, and R are the heating rate, the exothermic peak temperature, the pre-exponential factor, and the ideal gas constant (8.314 J/K·mol), respectively. As shown in Figure 1, all specimens displayed a single exothermic peak, and the T_p_ values moved towards a great temperature with the increasing β. It might be due to that the a lower β offers more time to proceed the curing process. Furthermore, the T_p_ and E_a_ values reduced with the increasing FATP content in case of the same β, indicating that the FATP improved the reactivity of the EP. The improved ring-opening reactivity contributed to both the polarization of epoxy ring, which originated from the hydrogen bond between −NH in the FATP and the epoxy group in the EP and the catalysis of the –N=N group from the tetra-azole of the FATP [44].

### 3.3. Flame Retardancy of the FREPs

The UL-94 rating and the LOI values were measured, and the corresponding data are summarized in Table 3. Obviously, the pristine EP was readily flammable and could not be self-extinguished once ignited, giving an LOI value of 25% and no UL-94 rating. It is satisfactory that the flame retardancy of FREPs was improved significantly with the loading of the FATP. In the case of the 2% FATP content, FREP-2 presented an LOI value of 32.5% and passed the UL-94 V-1 testing. With the FATP loading level up to 4% FATP (phosphorus content of 0.33%), the FRP-4 sample achieved an LOI value of 35% and a UL-94 V-0 rating. It demonstrated that the FATP had a better flame-retardant efficiency on the EP matrix.

### 3.4. Thermal Stability of the FREPs

Thermogravimetric analysis (TGA) was measured under N_2_ to assess the effect of the FATP on the thermal decomposition behavior of the FREPs. Appendix A and Table 4 show the resultant curves and data, respectively. As seen, the FATP presented a two-stage decomposition process. The first-stage occurred at 240–300 °C due to the thermal cracking of the tetrazole to release NH_3_ and trinitrides, and the second-stage ranging from 400 °C to 500 °C was attributed to the thermal cracking of phosphine group. Although the FATP gave a lower T_5%_ of 243.5 °C, its char residue at 700 °C was up to 33.2%, revealing that the FATP had a superior charring capacity. As seen, the pure EP and the FREPs displayed similar single-stage thermal decomposition processes in the range of 350–550 °C. It was also found that the T_5%_ values of EP/FATP-2 and EP/FATP-4 were reduced by 19.5 °C and 29.4 °C, respectively, as comparison with the pristine EP. It reflected that the FATP with a lower T_5%_ value accelerated the early degradation of the EP matrix. Noteworthy, the increasing FATP contents resulted in a lower R_max_ and enhanced the CR_700_ values of the FREPs. It might be owing to the fact that the FATP decomposed early at low temperature and produced some nonflammable gases, which restrained the further burning of the EP. Additionally, the char-forming capacity was improved by the reaction between the Furan ring in the FATP and the degradation products of the EP.

### 3.5. Analysis of Combustion Behaviors

The cone calorimeter test (CCT) is generally used to simulate the burning behaviors in a real fire, since it provides some characteristic parameters during the combustion [45,46]. Figure 2 illustrates the variations of the heat release rate (HRR), the total heat release (THR), the smoke release rate (SPR), and the total smoke release (TSP) with the time, and some crucial data are listed in Table 5.

As compared to the pristine EP, the FREPs presented lower values of time to ignition (TTI), e.g., 111 s for EP > 96 s for EP/FATP-2 > 85 s for EP/FATP-4. It is probably due to the fact that FATP with lower T_5%_ facilitated the early decomposition of the EP matrix.

The THR and the peak of the HRR (the PHRR) are important parameters to characterize the combustion intensity. The pure EP burned intensely and gave the highest values of the THR (95.7 MJ/m^2^) and the PHRR (1229.3 kW/m^2^). Clearly, the values of both the THR and the PHRR tended to decrease with the incorporation of the FATP. For example, the FREP-4 sample achieved the lowest values of the THR (86.7 MJ/m^2^) and the PHRR (1059.3 kW/m^2^), which were decreased by 13.8% and 9.4% by contrast with the pristine EP, respectively. It confirmed that the incorporation of the FATP suppresses effectively the heat release during burning.

Since the excess toxic smoke causes a mortal threat to victims in a fire, the TSP is a crucial parameter to assess fire safety of polymeric substances [47]. The pure EP displayed a higher TSP value of 139.7 MJ/m^2^. By contrast, the corresponding value of EP/FATP-4 was reduced by 35.9%, indicating that the FATP had better smoke suppression.

The average effective heat combustion (av-EHC) is defined as the ratio of the HRR to the mass loss rate (MLR) and often used to assess the combustion degree of volatiles in the gas phase [48]. The av-EHC values of the pure EP, FREP-2, and FREP-4 were achieved to be 24.47, 23.96, and 23.56 MJ/kg, respectively. It suggests that the incomplete combustion of volatiles occurs with the incorporation of the FATP. In contrast to the neat EP, the greater av-COY/av-CO_2_Y values of the FREPs further demonstrated the occurrence of the incomplete combustion. The above-mentioned phenomena are probably attributed to the fact that the phosphorus-containing species, which are produced from the degradation of the FATP, could quench combustible $H^{.}$ and ${\mathrm{OH}}^{.}$ during combustion, and the combustion chain reaction is thus interrupted [49,50]. What is more, the char residues of FREP-2 (15.5%) and FREP-4 (17.3%) were far more than that of the neat EP (9.4%). It indicates that the FATP promotes the formation of char residues which can isolate the transition of oxygen and flammable gases; it hence plays a flame-retardant effect in the condensed phase.

### 3.6. Characterization of Char Residues

Figure 3 shows the digital graphs and SEM micrographs of the char residues after CCTs for all samples. As seen in Figure 3a_1_,a_2_, the neat EP had fewer char residues with an expansion height of 1.6 cm. With the incorporation of the FATP, much more and intact char residues were left after CCTs, with the expansion heights of 2.5 cm for FREP-2 and 2.8 cm for FREP-4, respectively. It is mainly due to the fact that the dehydration and carbonization of EP is facilitated by phosphoric acid which are produced from the decomposition of FATP, and the char residue thus forms readily [51]. Furthermore, the char residues of the neat EP showed many holes and some cracks (Figure 3a_3_), which were conductive to the transmission of the heat and flammable gases from the inner EP matrix to the exterior environment. By contrast, a few voids were found in the char residues of FREP-2 (Figure 3b_3_). With the increasing FATP content, the char residues of FREP-4 became continuous and compact (Figure 3c_3_), which is in favor to suppress effectively the transmission of the heat and combustible volatiles.

It is well accepted that a Raman spectrometer was applied to evaluate the graphitization degree of carbon materials. As illustrated in the Raman spectra (Appendix A), two strong signals appeared at 1351 cm^−1^ and 1582 cm^−1^, which contributed to the D band representing disordered carbon and the G band representing graphitized carbon, respectively. The area ratio (I_D_/I_G_) of the D band to the G band is usually adapted to characterize the graphitization degree of carbon materials, and a lower I_D_/I_G_ value means a higher graphitization degree [52]. The calculated I_D_/I_G_ values showed a downtrend with the increasing FATP contents, that is, 2.64 for the pure EP > 2.62 for the EP/FATP-2 > 2.47 for the EP/FATP-4. The result confirmed that the incorporation of the FATP results in an enhancement of the graphitization degree, thus constituting readily a stable and compact carbon layer.

XPS was further applied to analyze the specific elemental composition and the valence state. Appendix A depicts the XPS spectra of the pure EP and FREP-4, and the specific elemental compositions are listed in Table 6. In addition to C, N, and O presenting in the char residues of the pure EP, an extra P element appeared in the char residues of FREP-4. As calculated, the C/O value of FREP-4 (6.53) was far greater than that of the neat EP (1.81). It indicated that the FATP had the char-forming capacity of the EP matrix and enhanced the resistance of the char to thermal oxidization, thus having a flame-retardant effect in the condensed phase.

Appendix A shows the C_1s_, O_1s_, N_1s_, and P_2p_ spectra of the FAEP-4 specimen. As seen in the C_1s_ spectrum, the appearance of three signals means that the carbon in char residues had three valence states. The peak at 284.7 eV was assigned to the C−C and C−H bonds in aliphatic and aromatic fragments, the one at 286.2 eV was attributed to the C−O−C and P−O−C bonds, and the one at 288.2 eV belonged to the C=O bond. The above bonds might be contributed to the carbonization and dehydration of the EP matrix during combustion. With respective to the O_1s_ spectrum, the peak at 531.3 eV belonged to the C=O and P=O bonds, and the one at 533.1 eV was ascribed to P−O−, C−O−C, or −O− in the P−O−P bond. As for the N_1s_ spectrum, the signals at 398.5 eV and 400.4 eV were attributed to the C−N bond and the N=N bond in the tetra-azole [53,54]. The P_2p_ spectrum presented two signals at 133.6 eV and 134.7 eV, which were ascribed to the P−O−C and P=O bonds, respectively. The results revealed that the FATP was decomposed to produce phosphorous oxides during combustion, which can catalyze the carbonization and dehydration of the EP matrix, and thus plays flame retardant roles in the condensed phase.

### 3.7. Gaseous Products Analysis

To better analyze the impact of the FATP in the gas phase, the gaseous volatiles released during the thermal decomposition of the pure EP and FREP-4 were detected via TG-FTIR. Figure 4 depicts the TG-FTIR spectra of the gases at different temperatures and the overall 3D TG-FTIR spectra. The main pyrolysis gases during the thermal decomposition of the pure EP and EP/FATP-4 were similar, including H_2_O (3622–3681 cm^−1^), aliphatic compounds (2750–3142 cm^−1^), aromatic compounds (3032 cm^−1^, 1600 cm^−1^, 1512 cm^−1^, and 827 cm^−1^), bisphenol A (1259 cm^−1^, 1337 cm^−1^, and 1175 cm^−1^), and carbonyl compounds (1732 cm^−1^). By contrast, in the spectra of FREP-4, some new characteristic peaks appeared at 1604 cm^−1^, 1255 cm^−1^, and 1050 cm^−1^, which contributed to the stretching vibration of P−O−Ph, P−O, and P−O−C, respectively. Therefore, it can be speculated that the decomposition of FATP produced ·HPO and ·PO radicals, which subsequently captured the reactive radicals such as H∙ and OH∙ to break off the combustion chain reaction. Two other peaks at 2309 cm^−1^ (HN_3_) and 919 cm^−1^ (NH_3_) were also found. Additionally, it was observed that the incorporation of FATP decreased both the initial temperature of the gas release (385 °C for the pristine EP and 356 °C for FREP-4) and the peak intensity. Appendix A further shows the variations of the peak intensities of some representative flammable volatiles with the time. As seen, EP/FATP-4 gave a lower intensity than the pure EP, suggesting that the FATP plays flame-retardant roles in the gas phase through the suppression effect on the release of flammable volatiles.

### 3.8. Possible Flame-Retardant Mechanism

Considering the above analysis of the combustion behavior, gaseous pyrolysis products, and the char residues, a possible flame-retardant mechanism was proposed and presented in Figure 5**.** In the gas phase, •HPO and •PO free radicals, which result from the decomposition of the phosphorous-containing fragment in the FATP, act as quenchers to capture the reactive groups such as •OH and •H. The chain reaction during the combustion is thus interrupted. Meanwhile, some nonflammable gases including HN_3_ and NH_3_ are also released and can dilute flammable volatiles and take a portion of heat away. In the condensed phase, the phosphorous-containing group in the FATP is decomposed into phosphoric acid, which promotes the catalytic carbonization and dehydration of the EP matrix. It, thereof, promotes the formation of a dense and expendable carbon layer in the surface of the EP matrix, which can hinder the transmission of the flammable gases and heat between the exterior environment and inner EP matrix. It is a synergistic effect among phosphaphenanthrene, furan, and tetra-azole, which means a better diphasic flame-retardant mechanism occurs; superior fire safety is thus achieved for FREPs.

### 3.9. Thermal and Mechanical Properties

The glass transition temperatures ($T_{g}$) of the neat EP and FREPs were tested via DSC. The corresponding DSC curves are illustrated in Appendix A, and the T_g_ values and the specific heat capacity values (ΔC_p_) are summarized in Table 7. As seen, the T_g_ values of FREPs were lower than that of the pristine EP and showed a downward trend with the increasing FATP content. This phenomenon might contribute to the reduction of the crosslinking density of FREPs upon the loading of the FATP. Conversely, the ΔC_p_ values showed an upward trend with the increasing FATP contents. It can be deduced that much more hydrogen bonds were constructed between −NH in the FATP and the epoxy group in the EP.

The dynamical mechanical performances were tested according to DMA; the variations of the storage modulus ($E^{\prime }$*)* and the loss tangent (tanδ) with the temperature are illustrated in Appendix A, in which the temperature at the peak of tanδ is defined as $T_{g}$. The crosslinking density ($V_{e}$) can be achieved according to the rubber elasticity theory with the equation $V_{e}=E^{\prime }/3RT$ [55], where $E^{\prime }$ and R donate the storage modulus at temperature of $T_{g}+40\;{}^{\circ}C$ and the gas constant (8.314 J/kmol), respectively. The values of $T_{g}$, $E^{\prime }$ at 50 °C, $E^{\prime }$ at $T_{g}+40\;{}^{\circ}C$, and $V_{e}$ are presented in Table 8. It was found that the FREPs gave lower $V_{e}$ values than that of the pristine EP and the $T_{g}$ of the FREPs was reduced accordingly [56]. The result is in agreement with that of the DSC. What we especially pointed out is that the $E^{\prime }$ value depends on both the temperature and the FATP contents. In the case of the temperature below $T_{g}$, for instance 50 °C, the $E^{\prime }$ values of the FREPs were higher than that of the pristine EP and displayed an upward trend with the increasing FATP content. It contributed to the construction of much more hydrogen bonds and the introduction of the rigid phosphaphenanthrene, furan, and tetra-azole in the FATP. At the temperature of $T_{g}+40\;{}^{\circ}C$, a contrary tendency of the $E^{\prime }$ against the FATP contents was observed. Specifically, the $E^{\prime }$ value was decreased from 42.4 MPa for the pure EP to 37.7 MPa for FREP-2 and 33.8 MPa for FREP-4. This phenomenon contributed to the fact that the negative effect of the reduced crosslinking density on the stiffness predominates over the positive effect of hydrogen bonds and rigid groups when polymer materials are in the rubbery state.

The effects of the FATP content on the mechanical properties of the FREPs were further evaluated through tensile and flexural testing, with results listed in Table 9. The pristine EP presented a tensile strength of 72.2 MPa and a flexural strength of 126.1 MPa. In the case of the 4 wt.% FATP loading, the corresponding values of EP/FATP-4 were increased to 74.8 MPa and 135.5 MPa, with an increase of 3.6% and 7.5% by comparison with the neat EP, respectively. Additionally, the incorporation of the FATP with rigid chain structure resulted in enhancements of the elastic and flexural moduli, which were similar with $E^{\prime }$ results in the temperature below $T_{g}$.

### 3.10. Transparency Analysis

EP thermosets with better transparency are generally used as optical materials such as LED and transparent coatings. It is well-known that the improvement of the flame retardancy is obtained at the expenses of the transparency of the EP, which is critical to the expended application of Ethe P. Therefore, the effect of the FATP on the optical transmittance of FREPs was explored. Figure 6 presents the curves of the transmittance against the wavelength in the range of 200–800 nm and optical images. Obviously, the transmittance (800 nm) of the EP was 89.3%, and the corresponding values of FREPs were almost the same, namely, 89.1% for EP/FATP-2 and 88.3% for FREP-4. Its better transparency allowed the emblems of Changzhou University to be clearly visible under all samples. The results demonstrated that the incorporation of the FATP had no negative roles on the transparency of EP thermosets. The maintenance of the better transparency contributed to the fact that the above-mentioned hydrogen bonds enhanced the compatibility between the FATP and the EP, thus leading to a uniform dispersion of the former within the later matrix. Moreover, UV-B and even UV-A were absorbed completely upon the introduction of FATP, indicating that the FREPs possessed better ultraviolet (UV) shielding capability. This specificity of FREPs is due to the fact that both the aromatic ring and the phosphorous-containing groups had the UV absorbability.

## 4. Conclusions

A nitrogen-rich DOPO-based derivate (FATP) was successfully synthesized by the one-pot method using 2-furaldehyde, 5-amino-1H-tetrazole, and DOPO as raw materials. FREPs were further prepared by curing the mixture of the EP and the FATP in the presence of the DDM as a curing agent. The analysis of the curing behavior showed that the FATP improved the reactivity of the EP and promoted the curing reaction. The TGA results confirmed that the incorporation of the FATP decreased T_5%_ and T_max_ as well as the maximum decomposition rate. When 4% FATP (0.33 wt.% phosphorus content) was introduced, the FREP-4 specimen achieved a UL-94 V-0 rating and an LOI value of 35%. Meanwhile, the EP/FATP-4 gave the lowest values of the THR (86.7 MJ/m^2^), the PHRR (1059.3 kW/m^2^), and the TSP (139.7 MJ/m^2^), with the reductions of 13.8%, 9.4%, and 35.9% as compared with the neat EP, respectively. The analysis of the pyrolysis gases revealed that the FATP played flame-retardant roles in the gas phase through the release of •HPO/•PO free radicals quenching reactive groups such as •OH and •H and some nonflammable gases (HN_3_ and NH_3_) diluting combustible volatiles. The subsequent characterization (SEM, Raman spectroscopy, and XPS) of char residues confirmed that the FATP promoted the formation of continuous and compact carbon layers with a greater graphitization degree, which acted as a physical barrier to inhibit the transmission of flammable gases and the heat between the exterior environment and the inner EP matrix, thus playing a flame-retardant effect in the condensed phase. The DSC and DMA results revealed that the storage modulus at 50 °C was increased although the Tg and Ve declined with the incorporation of the FATP. Compared with the pristine EP, FREP-4 kept almost the same transmittance while slightly enhancing mechanical properties. In summary, the FREPs not only possessed better flame retardancy, fire safety, and extra UV shielding capabilities, but also kept high transparency and mechanical strength, thus having promising applications in the fields of advanced optical technology.

## Data Availability

Not applicable.

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
