# Peer review of "A Nitrogen-Rich DOPO-Based Derivate for Increasing Fire Resistance of Epoxy Resin with Comparable Transparency"

_materials, 2023, doi:10.3390/ma16020519_

Round 1
Reviewer 1 Report
Authors have synthesized a nitrogen-rich DOPO based compound to be incorporated into an epoxy resin with the aim of prepare a flame retardant epoxy resin. They present a quite complete characterization via various techniques. It is practically interesting, however there is no novelty in the study.
Since the manuscript make reference to 9 figures in the supporting information and this SI has not been uploaded and has not been provided to me I am anable to accept the manuscript for publication.
Moreover, another minor questions should be addressed before accepting the manuscript for publication
- In the 3.1 Structural characterization of FATP (line 150) “The chemical structure of FTIR was characterized via FTIR” should be corrected.
- Line 212; Maybe CY700 should be replaced by CR700
- Section 3.9 Thermal and mechanical properties. Please indicate the heating rate used during the DSC run and the parameters (Hz,..) used for the DMA measurements.
Author Response
Authors have synthesized a nitrogen-rich DOPO based compound to be incorporated into an epoxy resin with the aim of prepare a flame retardant epoxy resin. They present a quite complete characterization via various techniques. It is practically interesting, however there is no novelty in the study.
Reply: Thanks for your comments. In our opinions, the highlights of the present work are as follows: (1) A P/N-containing compound (FATP) was synthesized using a bio-based 2-furaldehyde, a tetrazole with nitrogen-rich heterocycle, and DOPO as raw materials. (2) The incorporation of DOVNPT endows EP with simultaneously better flame retardancy and identical transparency and slightly enhavced mechanical strength as compared to the pure EP.
Since the manuscript make reference to 9 figures in the supporting information and this SI has not been uploaded and has not been provided to me I am anable to accept the manuscript for publication.
Reply: Sorry for the mistake, we have uploaded the supporting information online.
Q1:In the 3.1 Structural characterization of FATP (line 150) “The chemical structure of FTIR was characterized via FTIR” should be corrected.
Reply:We thank the reviewer for this valuable suggestion. These terms have been revised accordingly.
Q2:Line 212; Maybe CY700 should be replaced by CR700
Reply:We agree the reviewer’s suggestions thoroughly, and revised the manuscript accordingly.
Q3: Section 3.9 Thermal and mechanical properties. Please indicate the heating rate used during the DSC run and the parameters (Hz.) used for the DMA measurements.
Reply: Thanks for the reviewer's comments, I have revised the manuscript as follows: Different Scanning Calorimerter (DSC) was employed to investigate Tg of the pure epoxy resin and flame retardant EP blends. Samples need to be removed from thermal history.The samples were heated from 30 to 200 ℃ with a heating rate of 10 ℃/min. The mass of all the samples was ca. 10 mg. Dynamic mechanical analysis (DMA) was carried out on the Perkin Elmer DMA 8000 (PE, USA). The instrument needed to be calibrated once before use. Three point bending mode was used for sample measurement. The constant frequency was 1Hz and the amplitude was 20 μm. The prepressure was set to 0.02 N. The samples were heated from 30~250 ℃ at a heating rate of 10 ℃/min.

Reviewer 2 Report
Dear authors, first of all, I congratulate you for such an extensive study from the point of view of the characteristics followed. I went through your research with great interest and made a series of notes with observations in the body of the text. Mainly, a review of the translation of your text and several inherent corrections (figures, content) are needed. I also wrote down some questions, which refer to the course of the experiment (doesn't the different treatment speeds mean that another variable is introduced and the set of samples becomes more extensive?). Another uncertainty is related to the appearance and height of the residues, analyzed optically and with SEM. I look forward to your answers.

Author Response
Dear authors, first of all, I congratulate you for such an extensive study from the point of view of the characteristics followed. I went through your research with great interest and made a series of notes with observations in the body of the text. Mainly, a review of the translation of your text and several inherent corrections (figures, content) are needed. I also wrote down some questions, which refer to the course of the experiment (doesn't the different treatment speeds mean that another variable is introduced and the set of samples becomes more extensive?). Another uncertainty is related to the appearance and height of the residues, analyzed optically and with SEM. I look forward to your answers.
Reply: Thanks for your positive comments on our work. We have revised your comments in a point-by-point fashion as follows.
Q1: explain why the char residue is important for this study (in line 75)
Reply:We thank the reviewers for their valuable comments again. FATP plays an important role in the condensed phase, the amount of char residue increases when FATP is added. Dense, fluffy and char residue can better isolate the transfer of oxygen and heat.
Q2: explain DDM (in line 78)
Reply: We sincerely thank the reviewer for valuable comments to improve the manuscript. 4,4'- Diaminodiphenylmethane is DDM for short. DDM is a curing agent of epoxy resin in which the amino group can react with the epoxy group of the epoxy resin. For epoxy resin, whether as a binder or coating, curing agent is essential, otherwise epoxy resin cannot cure.
Q3: how transparent the solution could be, when a white powder was added? (in line 107)
Reply: We agree the reviewer’s suggestions thoroughly. When the flame retardant FATP is added to the epoxy resin, the secondary amino group of the FATP can react with the epoxy group of the epoxy resin, resulting in a yellow, transparent solution.
Q4: Those different heating rates become variable of the experiment? If so, how many samples do you have after-all? (in line 130)
Reply:First of all, thank the reviewers for their comments. The adoption of different heating rate is to investigate the non-isothermal curing kinetics. At least three samples of the same formula were tested at a heating rate.
Q5: for all the samples of each test set the heights were similar? (in line 263)
Reply: We sincerely thank the reviewers for their comments. Yes, the thickness of the samples for CCT is fixed as 3 mm, and the expansion height of the char residues left after CCT was similar.
Q6: It also seems that the surface of the EP has small and many holes, while the surface of the FATP-4 has some big holes. How you can explain that? (in line 272)
Reply: We sincerely thank the reviewers for their comments. The white part is the tinfoil paper at the bottom (Figure 3 a2), indicating that few char residues were left after CCT. As seen from the microstructure (Figure 3 c2), the small holes are connected together, while the surface of FATP-4 is compact and continuous.
Q7: on which device the XPS was recorded? (in line 288)
Reply: We sincerely thank the reviewers for their comments. The relevant test characterization instruments for XPS have been added in Chapter 2.4.
Round 2
Reviewer 1 Report
The authors have addressed the comments raised satisfactorily
Author Response
Appreciate greatly for your kind comments.
Reviewer 2 Report
Dear authors, thank you for the corrections. Yet, there are still some issues that you may take into consideration:
- I don`t find the Chapter 2.4 for the ``characterization instruments for XPS``
- I found in the revised text Figures S4, S5 and S6 - line 220, 230 and 237 - please correct
Author Response
Q1 I don`t find the Chapter 2.4 for the ``characterization instruments for XPS``
Reply: Thank for your comments. The specific characterization method and instruments were described in the Supporting information.
Q2 I found in the revised text Figures S4, S5 and S6 - line 220, 230 and 237 - please correct.
Reply: Thanks a lot. I checked the manuscript carefully and all supporting Figures are removed from the text.